# The Impact of Antimicrobial Therapy on the Development of Microbiota in Infants

**DOI:** 10.3390/antibiotics14121245

**Published:** 2025-12-09

**Authors:** Tatiana Priputnevich, Pavel Denisov, Ksenia Zhigalova, Vera Muravieva, Natalia Shabanova, Alexey Gordeev, Viktor Zubkov, Bayr Bembeeva, Elena Isaeva, Anastasia Nikolaeva, Gennady Sukhikh

**Affiliations:** 1National Medical Research Center of Obstetrics, Gynecology and Perinatology Named After Academician V.I. Kulakov, Ministry of Health of the Russian Federation, 117997 Moscow, Russia; 2Department of Microbiology and Virology, Institute of Preventive Medicine Named After Z.P. Solovyov, Pirogov Russian National Research Medical University (RNRMU), 117513 Moscow, Russia; 3Federal State Budgetary Educational Institution of Further Professional Education “Russian Medical Academy of Continuous Professional Education”, Ministry of Healthcare of the Russian Federation, Department of Medical Microbiology Named After Academician Z.V. Ermolyeva, 125993 Moscow, Russia

**Keywords:** gut microbiota, qPCR, AB therapy, AMR, microbial diversity

## Abstract

***Background.*** The establishment and diversity of the gut microbiota during early childhood are fundamental for immune regulation and metabolic processes, with factors such as prematurity, delivery method, antibiotic treatment, and breastfeeding significantly impacting microbiome development and potential health outcomes. ***Objectives/Methods.*** This comparative study examined the gut microbiota composition in children aged 6–8 and 9–12 months, born via spontaneous labor at ≥38 weeks’ gestation, who either did not receive antibacterial therapy or required beta-lactam antibiotics. The composition of the colonic microbiota was analyzed in these fecal samples using a quantitative real-time PCR (qRT-PCR). ***Results.*** Significant differences in microbiota composition were observed between groups. Children treated with antibiotics exhibited a statistically significant reduction in alpha diversity indices (Shannon and Simpson), along with decreased colonization of key functionally important microorganisms, including obligate anaerobic bacteria such as *Faecalibacterium prausnitzii*, *Clostridium leptum*, *Bacteroides* spp., and metabolically active *Bifidobacteria* (*B. bifidum*, *B. breve*, *B. longum*). ***Conclusions.*** These microbiota alterations may adversely affect child health by diminishing microbial balance and functional potential, especially during this critical period of immune and metabolic development. The decline in anti-inflammatory, short-chain fatty acid-producing bacteria elevates the risk for allergic, atopic, dysbiotic, and metabolic conditions. Recognizing these impacts underscores the importance of strategies to supports microbiota restoration after antibiotic use, such as probiotics, prebiotics, and dietary interventions. Further research should focus on microbiota recovery dynamics to facilitate early intervention and optimize pediatric health outcomes. Overall, understanding antibiotic effects on gut microbiota can guide more judicious treatment approaches, reducing long-term health risks.

## 1. Introduction

The human gut microbiota (GM) constitutes a complex and dynamic microbial community composed of bacteria, fungi, and viruses. Collectively, the GM supports host health and modulates the immune system [1].

The human microbiota plays a crucial role in the development and maintenance of health from early stages of ontogenesis. Members of the microbiota determine the immune response and resistance to pathogens, as well as participate in the exchange of a wide range of micro- and macronutrients [2,3,4,5]. Studies have demonstrated the presence of microbial RNA, including lipopolysaccharides from bacteria, in amniotic fluid and umbilical cord blood of newborns, indicating an interaction between microbiota and the immune system even at very early gestational stages [6,7,8]. Also, from birth, “non-sterility” is confirmed by the presence of 16S rRNA (microbial ribosomal RNA) in the meconium of children born at various stages of gestation [6,7,8].

During the first weeks postnatally, the colonization of the gastrointestinal tract begins, characterized by a progressive increase in microbial diversity [9]. The initial phase (from birth to two weeks) is dominated by aerobic microorganisms such as streptococci, as well as representatives of the gut microbiota, including *Clostridium* and *Bacteroides*. Subsequently, depending on feeding methods, there is an increase in the abundance of *Bifidobacterium*, *Lactobacillus*, and other anaerobic strains. As the child transitions to solid foods, the microbiota becomes more diverse, although preterm infants and infants delivered via cesarean section exhibit notable differences in microbial composition [10,11]. In term neonates, initial colonization is primarily transmitted from maternal sources, including vaginal and gut microbiota, with dominance of *Lactobacillus*, *Bacteroides*, and *Bifidobacterium* [12,13].

The concept emphasizing the importance of early life microbial development, often referred to as the ‘first 1000 days’, suggests that during this critical period, the foundational functions of the microbiome are established, exerting significant influence on immune development, metabolism, and overall health [14,15,16]. This period involves the formation of key interactions between the microbiome, immune regulation, and metabolic processes.

Factors such as gestational age, mode of delivery, and early postnatal environmental conditions substantially impact the composition and diversity of the gut microbiota [17,18,19,20,21,22,23]. Preterm infants tend to have reduced microbial diversity with a predominance of potentially pathogenic bacteria such as *Enterobacter*, *Klebsiella*, and *Staphylococcus*, with these alterations persisting at least until age four [24]. In physiological conditions, healthy full-term infants primarily acquire aerobic bacteria and maternal vaginal microbiota members, including *Lactobacillus*, *Bacteroides*, and *Bifidobacterium*, which play a vital role in initial colonization and subsequent microbiome development [12]. In previous studies, we demonstrated that the most significant factor influencing colonization processes is gestational age. Cesarean section, which prevents natural colonization by maternal microbiota, substantially inhibits the normal development of the intestinal microbiota [25]. These findings highlight the importance of maternal microbiota transfer for early microbial succession.

An influential factor in shaping gut microbiota is antibiotic administration. Despite their essential role in therapeutic interventions and improving survival rates, antibiotics—particularly broad-spectrum agents—can induce long-term alterations in the microbiota, including dysbiosis, diarrhea, colitis, and the development of metabolic, immune, and neurological disorders such as obesity, diabetes mellitus, irritable bowel syndrome, and autism spectrum disorders [26,27]. The impact of antibiotics depends on their spectrum, route of administration, and duration; oral intake tends to produce the most pronounced effects.

Changes induced by antibiotics are associated with the development of various pathological states, impairing immune regulation and the functions of symbiotic microorganisms, adversely affecting digestion, vitamin synthesis, and host defense mechanisms [28,29]. Current research confirms that the microbiome participates in synthesizing bioactive compounds and microorganisms, positioning it as a promising target for developing innovative therapeutic approaches.

The selection of β-lactam antibiotics in our study is justified by their high prevalence in pediatric practice, particularly among infants and young children. Epidemiological data indicate that these agents represent the most frequently prescribed class of antibiotics for treating bacterial infections of the respiratory tract, urinary tract infections, and other acute conditions [30].

Furthermore, β-lactam antibiotics possess broad-spectrum activity against both Gram-positive and Gram-negative bacteria and are readily accessible, which explains their preference in clinical settings. Consequently, examining their impact on the gut microbiota of children is especially relevant, since this class of antibiotics is considered among the safest options and is classified as first-line (empiric) therapy for early childhood infections. Their effects on the developing microbiome may have long-term clinical implications [31].

Penicillins, such as amoxicillin, are first-line agents for bacterial tonsillitis, pneumonia, and other infections in children, owing to their good tolerability and spectrum of activity. Cephalosporins, particularly third- and fourth-generation agents, carbapenems, are used in more severe cases or in cases of penicillin allergy, as well as for complicated infections such as sepsis or severe pneumonia [32]. Besides β-lactams, other antibiotic classes are widely used in pediatric practice, depending on the infection type, child’s age, and the suspected pathogen. Macrolides, such as azithromycin and erythromycin, are frequently employed in respiratory diseases and in cases of allergy or intolerance to penicillins. They are especially effective against atypical pathogens, including *Mycoplasma* and *Chlamydia*, making them an important group in the treatment of pediatric infections [33].

For severe infections caused by Gram-negative bacteria, aminoglycosides such as amikacin and netilmicin are utilized [34]. Given their increased toxicity risks to the kidneys and auditory system, especially in neonates and infants, their use requires careful dose management and blood level monitoring. Considering the high incidence of urinary system pathologies in our cohort, this group of antibiotics was not applied in our study.

Particularly significant is the influence of antibiotics on microbiome formation during a child’s first year of life. It is hypothesized that reducing or altering antibiotic-resistant strains may contribute to the development of chronic diseases and neurological disorders, including autism spectrum disorder [35]. These factors underscore the need for novel strategies to preserve and restore normal microbiota after antibiotic use.

The study of microbiome development in early childhood and its consequences under antimicrobial exposure is highly relevant for improving preventive and therapeutic strategies for pediatric diseases. A thorough understanding of the mechanisms of microbiota alterations and their influencing factors will facilitate the development of effective approaches to maintain health in early life and reduce future disease risks.

## 2. Materials and Methods

This is a cross-sectional study conducted at the National Medical Research Center of Obstetrics, Gynecology, and Perinatology named after Academician V.I. Kulakov of the Ministry of Health of the Russian Federation. The study included 60 infants, divided into two groups:Group I: 30 healthy infants aged 6–12 months, born full-term (38–40 weeks of gestation), delivered vaginally, and who had not received antibiotic therapy.Group II: 30 infants aged 6–12 months, born full-term (38–40 weeks of gestation), delivered vaginally, who had received antibiotic therapy with β-lactam antibiotics (penicillins, cephalosporins, carbapenems). Most children in this group were born at the National Medical Research Center of Obstetrics, Gynecology, and Perinatology named after V.I. Kulakov, with five children born in various other medical institutions across Russia.

Children in Group II required antibiotics for various reasons, including congenital pneumonia, urinary tract infections, and post-surgical interventions for congenital defects unrelated to the gastrointestinal tract (such as megaureter, hydronephrosis, multicystic kidney, lung malformation, inguinal hernia, external genital cyst, mediastinal tumors). The majority received antibiotics postnatally during the postoperative period (20 children, 66.7%), two children received antibiotics at three months of age due to suspected pyelonephritis, and eight children reportedly received penicillin-based medications on an outpatient basis, according to parental reports.

The groups were comparable in terms of feeding:In both groups, 16 children in Group I and 14 in Group II were breastfed.14 children in Group I (46.6%) and 16 in Group II (53.3%) were fed artificially (formula-fed).

Additionally, the distribution of complementary feeding was similar:28 children in Group I (93.3%) and 25 in Group II (83.3%) received complementary foods.Two children in Group I and five in Group II did not.

All children included in the study were under observation at the Pediatric Consultative Department of the National Medical Research Center of Obstetrics, Gynecology, and Perinatology named after V.I. Kulakov of the Ministry of Health of Russia. They were either observed from birth or upon admission between 6 and 12 months of age for delayed surgical correction of congenital defects. Importantly, all participants had not received antibiotics in the month prior to the start of the study (Table 1 and Table 2).

Fecal samples were collected after a natural bowel movement and placed into sterile containers with proper labeling. In the study, the composition of the colonic microbiota was analyzed in these fecal samples using a quantitative real-time PCR (qRT-PCR) method. For this, a registered multiplex reagent kit in the Russian Federation was employed, containing primers for detecting DNA of gut-associated microorganisms, including those that are difficult to culture (DNA-Technology, Moscow, Russia). The primers targeted representatives from the following phyla: Bacillota (Firmicutes), Pseudomonadota (Proteobacteria), Bacteroidota (Bacteroidetes), Actinomycetota (Actinobacteria), Fusobacteriota (Fusobacteria), Verrucomicrobiota (Verrucomicrobia), and Euryarchaeota. Additionally, the kit included primers for detecting fungi of the genus *Candida*, genes responsible for methicillin resistance (mecA) in *Staphylococcus* spp., toxins A and B (tcdA, tcdB) of *Clostridioides difficile*, and the srr2 gene associated with invasiveness in *Streptococcus agalactiae*.

### 2.1. PCR Procedure

Nucleic acids were extracted from the biological material using the “Proba NK-plus” reagent kit (DNA-Technology, Moscow, Russia). The gut microbiota was analyzed using the ENTEROFLOR Kiddy Kit (DNA-Technology, Moscow, Russia) in the molecular microbiology laboratory of the Institute of Microbiology, Antimicrobial Therapy, and Epidemiology at the National Medical Research Center for Obstetrics, Gynecology, and Perinatology named after Academician V.I. Kulakov of the Ministry of Health of the Russian Federation.

DNA extraction from the test material was performed according to the manufacturer’s instructions for the respective reagent kit. Quality control of the extracted DNA was carried out during PCR using an internal control system (ICS). Detection was performed using the Realtime DNA Amplifier “DTprime” (DNA-Technology, Moscow, Russia).

Results are expressed as the decimal logarithm (lg) of the number of genome equivalents (GEs) of microorganisms, pathogenicity, or resistance factors per 1.0 g of clinical material (GEs/g).

### 2.2. Statistical Analysis and Visualization

Analysis was performed using OriginLab Pro 2021 (version 9.8.0.200, OriginLab Corporation, Northampton, MA, USA). Normality of distribution was tested using the Kolmogorov–Smirnov test. Data that followed a normal distribution were presented as mean ± standard deviation (SD), and comparisons were made using Student’s *t*-test. In other cases, data were presented as median with interquartile range (Me (Q25–Q75)), and the Mann–Whitney U test was used. Percentages (%) were calculated for qualitative data.

### 2.3. Biodiversity Indices

Species diversity (SDI) is a key characteristic of a community, reflecting its complexity. It is associated with the stability of the biocenosis and can serve as an indicator of various ecological aspects, such as the level of disturbance, energy flow, environmental stability, and more [36].

A decrease in species diversity indicates simplification of the community’s species structure and disruption of the balance between species by abundance. SDI includes two components: species richness (number of species) and evenness (distribution uniformity across species).

Quantitative measures of SDI include various diversity indices. Over 30 different indices are used to assess different aspects of biodiversity. In this study, two indices that consider both components—species richness and evenness—were employed (see Table 3). Specifically, the Simpson and Shannon indices were used for calculating taxonomic diversity. Differences in the statistical values were considered significant at *p* < 0.05.

The Simpson index is more sensitive to changes in the abundance of the most dominant species, whereas the Shannon index is more responsive to changes in the abundance of rare species. Therefore, the Simpson index is preferable when the community is mainly characterized by dominant species, while the Shannon index is recommended when a comprehensive assessment of the entire species composition, including rare species, is required.

## 3. Results

In total, PCR analysis was conducted on fecal samples from 30 healthy children not receiving antibiotic therapy (ABT) and 30 children undergoing ABT. The results are presented in Figure 1. It was demonstrated that children with ABT showed a statistically significant decrease in biodiversity indices, as determined by the two-sample *t*-test: Shannon index (*p* = 0.03698) and Simpson index (*p* = 0.0182).

To evaluate the impact of antimicrobial therapy (ABT) on gut microbiota (GM) composition, a comparison was made of the qualitative and quantitative profiles of microorganisms between children not receiving ABT and those undergoing ABT. The results are shown in heatmaps (Figure 2). Additionally, a comparison of genome equivalents per gram of feces (GEs/g) was performed between the groups.

A more detailed analysis of the microbiota composition, focusing on the most significant microorganisms, is presented below (Figure 3). In the comparison of the quantitative component of the microbiota (GEs/g) between the studied groups of children, a statistically significant decrease was observed in the populations of *Clostridioides difficile* (two-sample *t*-test, *p* = 0.029), as well as obligate anaerobic microorganisms that produce short-chain fatty acids (SCFAs), such as *Faecalibacterium prausnitzii* (*p* = 0.017), the *Clostridium leptum* group (*p* = 0.046), *Bifidobacterium* spp. (*p* < 0.001), and *Bacteroides* spp. (*p* = 0.01178) in the group of children receiving antibiotics.

*Bifidobacteria* are primary representatives of a healthy gut microbiota in children during the first two years of life. These bacteria possess anti-inflammatory properties: they suppress the synthesis of certain interleukins, enhance immune responses, and protect the intestinal wall from damaging agents. *Bifidobacteria* participate in carbohydrate metabolism and produce short-chain fatty acids (SCFAs)—such as acetic acid, lactic acid, and formic acid—that lower the pH of the environment, creating unfavorable conditions for pathogenic microorganisms. Additionally, they have a beneficial effect on intestinal immunity. A reduction in *Bifidobacteria* populations is associated with an increased risk of developing inflammatory bowel diseases and nonspecific ulcerative colitis.

It is believed that *Bifidobacteria* can be classified into “infant” and “adult” types. The “infant” *Bifidobacteria* include *Bifidobacterium longum* subsp. *infantis*, *Bifidobacterium longum* subsp. *longum*, *Bifidobacterium breve*, and *Bifidobacterium bifidum*. These strains facilitate the complete digestion of breast milk oligosaccharides and contribute to the activation of the anti-inflammatory arm of the immune response. The “adult” group comprises species such as *Bifidobacterium catenulatum*, *Bifidobacterium adolescentis*, *Bifidobacterium dentium*, and *Bifidobacterium animalis*. When “adult” species predominate in children, there may be a pro-inflammatory activation of the immune system.

Regarding the detection frequency of metabolically active “infant” *Bifidobacterium* species, no statistically significant difference was observed (Table 4).

However, based on the quantitative indicators, a statistically significant reduction in colonization levels was observed for metabolically active “infant” *Bifidobacterium* species, including *Bifidobacterium bifidum* (*p* = 0.03551), *Bifidobacterium breve* (*p* = 0.01601), and *Bifidobacterium longum* subsp. *longum* (*p* < 0.001) (Figure 4).

Regarding the detection frequency of metabolically active “adult” *Bifidobacterium* species, no statistically significant differences were observed, similar to the “infant” species (Table 5). In the comparison of quantitative indicators for most metabolically active “adult” *Bifidobacterium* species, no significant differences were found, except for *Bifidobacterium catenulatum* (two-sample *t*-test, *p* = 0.00263) (Figure 5), where a decline in their numbers was observed in the group receiving antibiotics.

Among the most prevalent gut-associated microorganisms are members of the Bacillota phylum (Firmicutes). The evaluation of the detection frequency of Bacillota (Firmicutes) representatives revealed a statistically significant decrease for the group of microorganisms comprising *Dialister + Allisonella + Megasphaera + Veillonella* (96.7% in children without antibiotic therapy [No ABT] vs. 66.7% in children with ABT), as well as for the family Lachnospiraceae (90.0% vs. 66.7%, respectively) (Table 6).

The microbial group *Clostridium leptum* (cluster IV) includes species such as *C. leptum*, *C. sporosphaeroides*, *C. cellulosi*, and *F. prausnitzii*. Cluster IV, known as the *C. leptum* group, is considered a dominant component of fecal microbiota in adults, with its abundance approaching 16–25%. A characteristic feature of this group is its synergistic interaction with various gut microorganisms during the fermentation of undigested carbohydrate fibers. The production of short-chain fatty acids by these bacteria serves as a primary energy source for the colonic epithelium and plays a role in regulating intestinal epithelial function.

Notably, among microorganisms in this group, a statistically significant reduction in quantity was observed only for *F. prausnitzii* (*p* = 0.01664) and the *Clostridium leptum* group overall (*p* = 0.04547) (Figure 6, Table 6).

*F. prausnitzii* is the most frequently encountered and representative species within the *C. leptum* group. This species is notable for its significant butyrate production through carbohydrate fermentation and is believed to play an anti-inflammatory role in the gastrointestinal tract. An imbalance in gut microbiota composition—particularly a decrease in *Bifidobacterium* and members of the *C. leptum* group such as *F. prausnitzii*—can potentially lead to serious gastrointestinal diseases, including inflammatory bowel diseases (IBD) and ulcerative colitis [37].

It is known that *F. prausnitzii* helps prevent inflammation by inhibiting the nuclear factor kappa-light-chain-enhancer of activated B cells (NF-κB) pathway and the release of interleukin-8 (IL-8). Our study demonstrated that the detection prevalence of *F. prausnitzii* was higher in children in the AB group (*n* = 22, 73.3%) compared to children without AB (*n* = 12, 40.0%). However, the quantitative abundance of this microorganism was significantly decreased in the ABT group (*p* = 0.01664), in conjunction with reduced populations of *Bifidobacterium* spp. and the overall *C. leptum* group (Figure 6, Table 7).

Analysis of gut colonization by *F. prausnitzii* depending on age (6–8 months and 9–12 months) revealed that the abundance of this microorganism increases with age in both groups, regardless of whether they received antibiotic therapy (AB) (Table 8; Figure 7).

The family Lactobacillaceae, also classified within the phylum Bacillota (Firmicutes), consists of microorganisms that are among the first to colonize the newborn’s intestine. They contribute to establishing colonization resistance within the gut biotope, acidify the environment through lactic acid production, and create unfavorable conditions for pathogenic microorganisms. Compared to *Bifidobacteria*, their proportion is significantly lower—about 5% in children older than 3 months.

In our study, there was a tendency toward a decrease in the detection frequency of Lactobacillaceae (including lactic acid bacteria) in children with ABT (80.0%) compared to children without ABT (63.3%), although this difference was not statistically significant. When comparing the level of gut colonization by species of the Lactobacillaceae family, a similar trend toward reduction was observed in the ABT group (*p* = 0.16).

The phylum Bacteroidota (formerly Bacteroidetes) is the second-most abundant group in the gut microbiota. Members of this phylum play a crucial role in polysaccharide metabolism, serve as primary producers of anti-inflammatory short-chain fatty acids (SCFAs) in the intestine, and secrete metabolites involved in regulating immune and nervous system functions. Microorganisms belonging to Bacteroidota appear in the gut microbiota in significant quantities after the introduction of solid foods in children and are an essential component of the physiological microbiota. Their abundance is strongly influenced by dietary patterns; for example, it is usually reduced in vegetarians [38].

The most common genus within this group is Bacteroides. Numerous species within this genus have multiple roles, including providing protection against pathogens, supplying nutrients to other microorganisms in the gut, and participating in the secretion of antimicrobial toxins in a contact-independent manner. When interacting with target cells, these multifunctional substances release antimicrobial toxins and effectors that mediate inter-bacterial antagonism [39].

Assessment of the detection frequency of members of the Bacteroidota (Bacteroidetes) phylum, such as *Alistipes* spp., *Bacteroides* spp., *Butyricimonas* spp., *Parabacteroides* spp., and *Prevotella* spp., revealed no statistically significant differences (Table 8).

When comparing the populations of microorganisms within the Bacteroidota (formerly Bacteroidetes) group, a statistically significant decrease was observed only in the genus *Bacteroides* in children receiving ABT (two-sample *t*-test *p* = 0.01178) (Figure 8).

The group of conditionally pathogenic microbiota included the following indicators: *Clostridium difficile* cluster, *Enterococcus* spp., Erysipelotrichaceae, *E. coli*, Enterobacterales, Peptoniphilaceae, *Pseudomonas* spp., Fusobacteriaceae, *Clostridium perfringens* cluster, and *Staphylococcus* spp.

Regarding detection frequency, a statistically significant reduction was noted for *Clostridium perfringens* (73.3% in children without ABT vs. 43.3% with ABT), *E. coli* (93.3% vs. 66.7%), Erysipelotrichaceae (80.0% vs. 53.3%), and Peptoniphilaceae (40.0% vs. 16.7%) (Table 9).

A statistically significant decrease in quantity was observed only in the *Clostridioides difficile* group (two-sample *t*-test, *p* = 0.029) (Figure 9). Representatives of the genus *Pseudomonas* were detected only once in the group not receiving antimicrobial therapy.

*Clostridioides difficile* (*C. difficile*) is part of the normal microbiota of the gastrointestinal tract (GIT), primarily colonizing the colon and less frequently the small intestine. Its normal abundance does not exceed 0.01–0.001% of the total gut microbiota. In most newborns (15–70%), both toxigenic and non-toxigenic strains of *C. difficile* are present as part of the natural microbiota. Over time, a stable microbiota develops, dominated by commensal microorganisms that protect against toxigenic *C. difficile* strains through mechanisms such as competition for nutrients, production of short-chain fatty acids (SCFAs), and bacteriocins.

Under certain conditions—such as a reduction in microbial diversity—*C. difficile* can cause serious diseases. *C. difficile*-associated disease (CDAD) develops when pathogenic bacteria overgrow, and their toxins damage the intestinal mucosa, leading to inflammation of the colon wall and diarrhea. Severe inflammation of the colon, known as pseudomembranous colitis, results from epithelial damage and decreased blood flow, with the formation of fibrinous or purulent exudates on the mucosa.

*Clostridioides difficile* infection (CDI) primarily occurs when the normal microbiota is disrupted. Risk factors include immunosuppression and antibiotic use. During antimicrobial therapy, changes in the gut microbiota facilitate colonization of the colon by toxigenic *C. difficile*, which is considered the initial step in the development of infection.

In our study, the prevalence of *C. difficile* was 30% in both groups of children. Toxigenic strains were identified by detecting genes encoding toxins A and B (tcdA and tcdB). The proportion of toxigenic strains was 16% among children not receiving antibiotics and 20% among those treated with antibiotics.

The lack of significant differences between the groups in the detection rate of *C. difficile*, including in children on antibiotics, indicates that carriage of both toxigenic and non-toxigenic strains in children under 2 years old is common and does not pose a health threat. This likely relates to the absence of specific toxin receptors on enterocytes at this age.

Colonization of the GIT by conditionally pathogenic microorganisms, which are an essential part of the normal microbiota, such as *Escherichia coli* and *Enterococcus* spp., was observed in 93.3% and 100% of children in the group without antibiotics, respectively. In the antibiotic-treated group, these rates decreased to 66.7% and 86.7%, respectively. The quantitative levels of these microorganisms within the microbiota were similar across both groups.

*Staphylococcus aureus*, when present at levels below the threshold, can be detected during eubiosis (a balanced microbiota). However, disruption of microbiota homeostasis can delay its clearance and lead to infections of varying severity, including sepsis. The detection rate of *S. aureus* was 47.0% in children without antibiotics and 40.0% in children with antibiotics (Table 10). Microbial load levels did not differ significantly, with median counts not exceeding 4–5 log CFU/g (see Figure 10).

The presence of yeast fungi, including *Candida albicans*, is always assessed as part of the microbiota—specifically, the absolute quantity of yeast fungi. *C. albicans* is normally absent or detected in minimal amounts within the microbiota. Its presence may be associated with microbiota disruption or insufficient bacterial colonization of the intestine. This is especially relevant during antimicrobial therapy, which often leads to secondary candidiasis. In our study, the detection frequency and quantitative characteristics of yeast fungi of the genus *Candida* did not differ statistically between the groups of children receiving antibiotics and those not receiving antibiotics (Table 11, Figure 10).

### Multiple Regression Analysis

To evaluate the influence of factors such as antibiotic therapy and breastfeeding on gut microbiota composition, a multiple regression analysis was performed. The response variables comprised 36 factors, while the predictor variables included two binary factors: breastfeeding and antibiotic use. This methodology enabled the assessment of the statistical significance of each predictor in explaining variations in the microbiota, as well as the elucidation of potential associations between these factors. The results, presented below, highlight the significance levels of the predictors and contribute to a comprehensive understanding of the determinants influencing the development and homeostasis of the gut microbial ecosystem.

Regression analyses were conducted using the ‘statsmodels’ library [40], resulting in detailed outputs that include coefficient estimates, their statistical significance, and model quality metrics. Multivariate regression analysis revealed that antibiotic therapy (ABT) has a significant impact on multiple parameters of the gut microbiota. The most notable changes were observed in taxa such as *Bifidobacterium* spp., *Bifidobacterium catenulatum* ssp., *Dialister-Allisonella-Megasphaera-Veillonella*, *Coriobacteriia*, Lachnospiraceae, Erysipelotrichaceae, *E. coli*, Peptoniphilaceae, and *Clostridium perfringens.*

Several parameters exhibited *p*-values approaching the significance threshold (e.g., *Bacteroides* spp., Enterobacterales, Lactobacillaceae, Fusobacteriaceae), indicating a potential influence of antibiotics on these groups, although they did not attain formal statistical significance.

Regarding the effect of breastfeeding, several parameters demonstrated statistically significant changes. Notably, *Staphylococcus aureus*, *Faecalibacterium prausnitzii*, *Akkermansia muciniphila* and Peptoniphilaceae exhibited notable alterations in the microbiota of breastfed children. For other taxa, such as *Clostridioides difficile*, *Clostridium leptum*, *Bifidobacterium catenulatum* spp., *Bifidobacterium breve*, and *Streptococcus agalactiae*, the effects were close to the significance threshold. The results of the analysis are presented in Table 11. To review the comprehensive data from the OLS regression analysis, please refer to the Appendix A.

## 4. Discussion

Antibiotics have significantly contributed to fighting diseases and increasing human life expectancy, but their impact on the gut microbiome raises concerns. In particular, broad-spectrum antibiotics not only eliminate harmful pathogens but also destroy beneficial bacteria that are crucial for maintaining gut health and overall well-being [41].

The use of antibiotics in newborns during their early years remains a significant and current issue in pediatrics and microbiology. This period involves the active formation of the microbiota, which plays a crucial role in immune system development, metabolic processes, and physiological functions of the body [42]. Numerous studies confirm that improper or excessive use of antibiotics can disturb microbial communities in children’s intestines, causing dysbiosis and imbalances [43,44,45,46,47]. Such conditions are associated with an increased risk of developing allergies, asthma, diabetes mellitus type 1, and chronic diseases later in life [48,49,50,51,52]. These findings emphasize the importance of rational antibiotic use and control over their prescription.

Particularly vulnerable are newborns and preterm infants, whose microbiota development is especially difficult, increasing the likelihood of complications and diseases related to microbial imbalance. Excessive use of antibiotics promotes the development of resistant bacterial strains and has long-term negative effects on a child’s health [53,54]. Recently, significant attention has been directed towards developing and implementing methods such as probiotics and prebiotics to restore and support microbiota during antibiotic therapy. Modern research shows that proper use of these agents helps reduce the risk of dysbiosis and restore the balance in the gut [55].

Many studies have demonstrated that infants exposed to Intrapartum Antibiotic Prophylaxis (IAP) in the first weeks of life exhibit lower proportions of *Actinobacteria* and *Bacteroidetes* [56,57], high oral *Proteobacteria* levels [4], and lower levels of *Bifidobacteria* [58]. At 3 months, they show underrepresentation of *Bacteroides*, *Parabacteroides* and higher *Enterococcus* and *Clostridium* [59] as well as a higher abundance of Enterobacteriaceae [58], as compared to nonantibiotic-exposed infants. Our data also confirm changes in the representation of these taxon levels in children aged 6–12 months.

In our study, we aimed to exclude children with gastrointestinal pathologies, as these are a significant factor affecting the gut microbiota. We believe that other conditions influence the microbiota less clearly, although they can still affect it through various mechanisms. Nonetheless, the presence of any pathological conditions in children and their potential impact on gut microbiota from the first days of life remain limitations of our study.

Advances in diagnostic technologies, such as metagenomics, now allow for more precise assessment of microbiota status in individual patients, leading to more appropriate antibiotic prescribing and avoiding overuse. The effective application of these technologies not only reduces the prevalence of antibiotic-resistant strains but also enables more targeted therapy, which is especially important for newborns and immunocompromised children [43].

Overall, current approaches to treatment highlight the necessity of a comprehensive strategy, including prevention, probiotic use, timely diagnosis, and minimizing unnecessary antibiotic prescriptions.

Today, improving standards and guidelines for antibiotic use in children, alongside the integration of new diagnostic and therapeutic methods, remains crucial to ensuring safety and health for children in the long term.

An important aspect of our study was the identification of a correlation between antibiotic therapy and the reduction in key microbial groups crucial for maintaining gastrointestinal health. Particularly noteworthy is the significant decrease in populations of SCFA-producing bacteria such as *Faecalibacterium prausnitzii* and members of the *Clostridium leptum* group in children who received antibiotics. These microorganisms are involved not only in metabolism and immune system development but also in strengthening the intestinal barrier function.

Further research that could be beneficial includes monitoring the dynamics of microbiota recovery during the post-antibiotic period. This would help us understand how quickly and through what mechanisms the microbiota returns to its normal state, as well as evaluate the effectiveness of different correction methods—such as probiotics, prebiotics, and dietary interventions.

Another interesting direction is studying the potential long-term consequences of reduced microbial diversity in children, especially in the context of immune and metabolic disorders. This knowledge would support the development of more precise guidelines for antibiotic use in infants and help identify optimal strategies for supporting the microbiome.

Additional promising areas of research include investigating the role of specific microbial strains in the development of pediatric diseases and their potential use as targeted probiotic therapies. Exploring the impact of different antibiotic classes on microbiota composition and function can help inform more selective prescribing practices. Developing personalized approaches to microbiota restoration based on individual microbiome profiles is also a valuable direction. Examining the influence of diet and environmental factors on microbiota resilience and recovery after antibiotic treatment may provide insights into supporting microbiota health. Finally, assessing the relationship between early microbiota disruptions and later neurodevelopmental outcomes, such as cognitive and behavioral development, can help understand long-term effects. These research areas could significantly advance our understanding of microbiome dynamics in early childhood and improve clinical strategies for microbiota preservation.

## 5. Conclusions

Thus, the conducted comparative study of the microbiota composition in two groups of children aged 6–8 and 9–12 months, born via spontaneous labor (gestational age from 38 weeks), who did not receive antibacterial therapy, and children who required beta-lactam antibiotic treatment, revealed significant differences in microbiota composition.

Children who received antibiotics showed a statistically significant reduction in alpha diversity indices (Shannon and Simpson). The microbiota of children in this group exhibited decreased colonization levels of functionally important microorganisms, including obligate anaerobic bacteria that produce various biologically active substances, including short-chain fatty acids (SCFAs). A statistically significant decrease was observed in populations such as the *Clostridium leptum* group, including *Faecalibacterium prausnitzii*, bacteria of the *Bacteroides* genus, and metabolically active “child” species of *Bifidobacteria* (*Bifidobacterium bifidum*, *Bifidobacterium breve*, and *Bifidobacterium longum* subsp. *longum*). A trend toward reduced colonization levels was also noted among members of the *Lactobacillaceae* family, which includes lactic acid bacteria.

These microbiota changes caused by antibiotic therapy may significantly impact children’s health in the future. A decrease in alpha diversity indicates a loss of microbial balance and a reduction in the functional potential of the microbiota. This is especially critical in early childhood, a key period for immune system development and the foundation for subsequent metabolic and immune health. Obligate anaerobic bacteria such as *Faecalibacterium prausnitzii*, known for their anti-inflammatory properties and ability to produce SCFAs vital for nourishing epithelial cells of the large intestine and maintaining the intestinal barrier function, were notably decreased. Reduction in these bacteria and other key microbiota components can increase the risk of developing various conditions, including allergic reactions, atopic dermatitis, dysbiosis, and metabolic disturbances.

Although antibiotic treatment is necessary in cases of severe infections, early intervention requires a thoughtful approach with attention to potential long-term effects on gut microbiota composition. Therefore, it is important to explore methods of supporting and restoring the microbiota after antibiotic therapy, including probiotic and prebiotic interventions, as well as dietary recommendations promoting the growth and colonization of beneficial microorganisms. Additional research should focus on studying the dynamics of microbiota recovery during the post-antibiotic period and identifying factors that facilitate a quicker return to a normal microbiological balance.

In summary, a comprehensive understanding of the impact of antibacterial therapy on the gut microbiota in children will help optimize treatment approaches and minimize possible adverse effects, thus improving the quality and prognosis of child health.

## Figures and Tables

**Figure 1 antibiotics-14-01245-f001:**
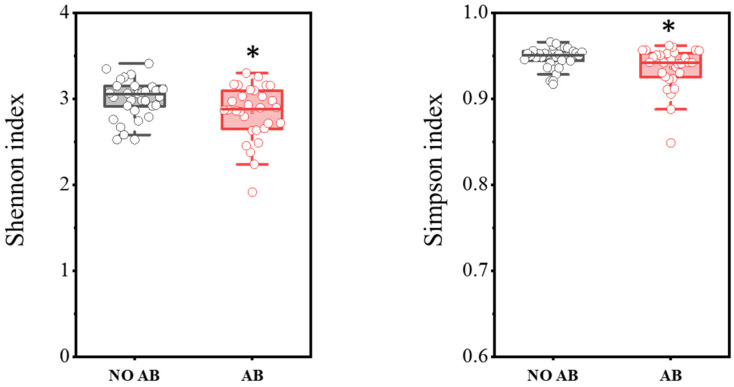
Comparison of biodiversity indices between the NO AB (healthy children without antibiotic therapy) and AB (children undergoing antibiotic therapy) groups. Two sample *t* Test, * *p* < 0.05.

**Figure 2 antibiotics-14-01245-f002:**
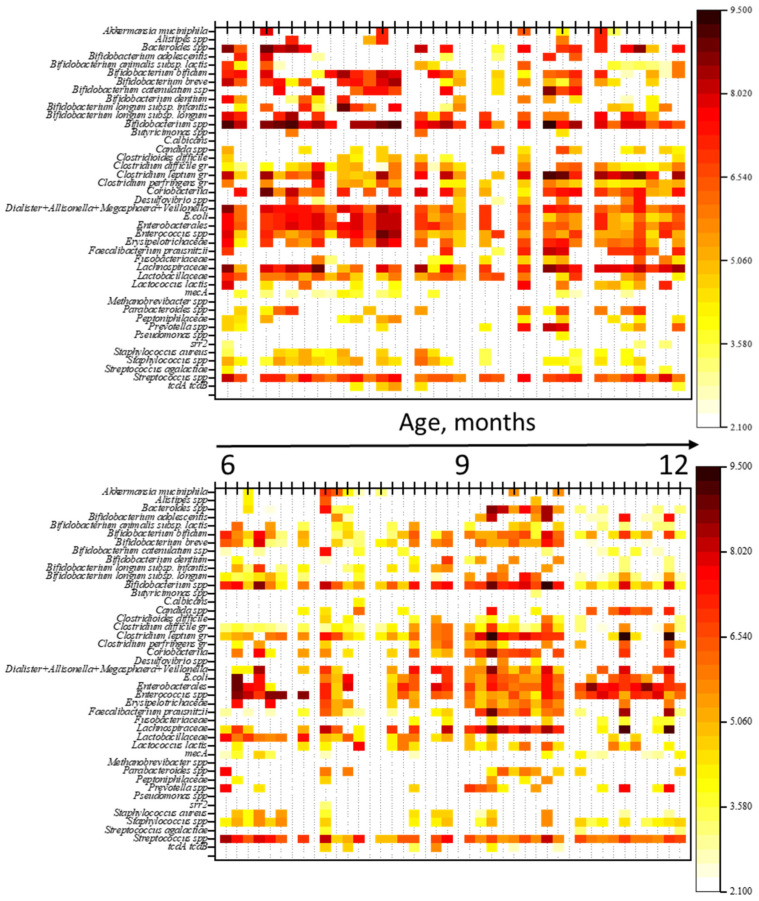
Heatmaps of gut microbiota profiles in children: the top heatmap represents the NO AB group (children without antibiotic therapy), and the bottom heatmap represents the AB group (children receiving antibiotic therapy). Note: The figure provides a graphical representation of the quantitative and qualitative composition of the gut microbiota in both groups of children. Each thin vertical line corresponds to an individual child’s microbiota profile, with the abundance of microorganisms in GEs/g (genome equivalents per gram) indicated by color.

**Figure 3 antibiotics-14-01245-f003:**
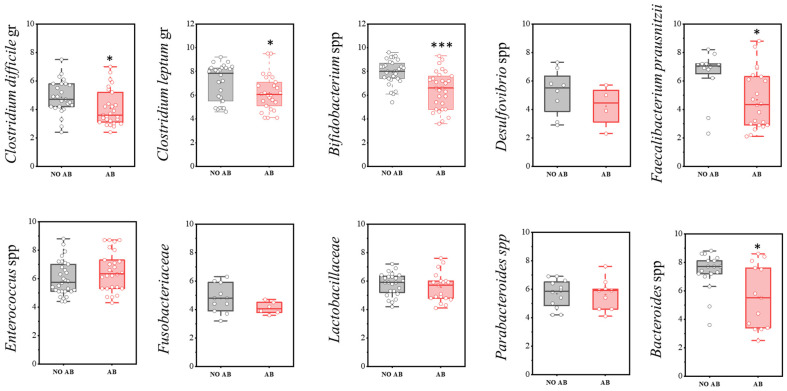
Analysis of the quantitative indicators (GEs/g) of main microbial groups identified by real-time PCR in children: AB group (receiving antibiotics) and NO AB group (not receiving antibiotics). Two sample *t* Test, *** *p* < 0.001, * *p* < 0.05.

**Figure 4 antibiotics-14-01245-f004:**
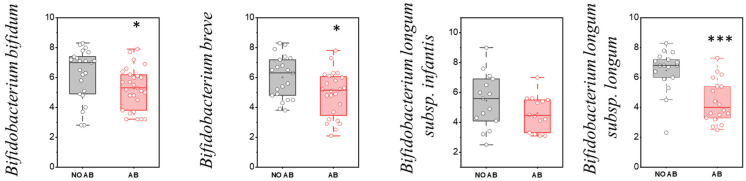
Analysis of quantitative indicators (GEs/g) of metabolically active “infant” *Bifidobacterium* spp. in children: without antibiotic therapy (No AB) and with antibiotic therapy (AB). Two sample *t* Test, *** *p* < 0.001, * *p* < 0.05.

**Figure 5 antibiotics-14-01245-f005:**
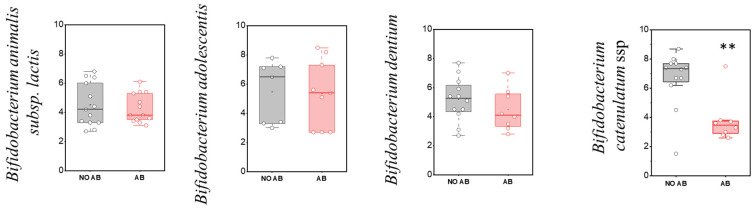
Analysis of quantitative indicators (GEs/g) of metabolically active “adult” *Bifidobacterium* spp. in children: without antibiotic therapy (no AB) and with antibiotic therapy (AB). Two sample *t* Test, ** *p* < 0.01.

**Figure 6 antibiotics-14-01245-f006:**
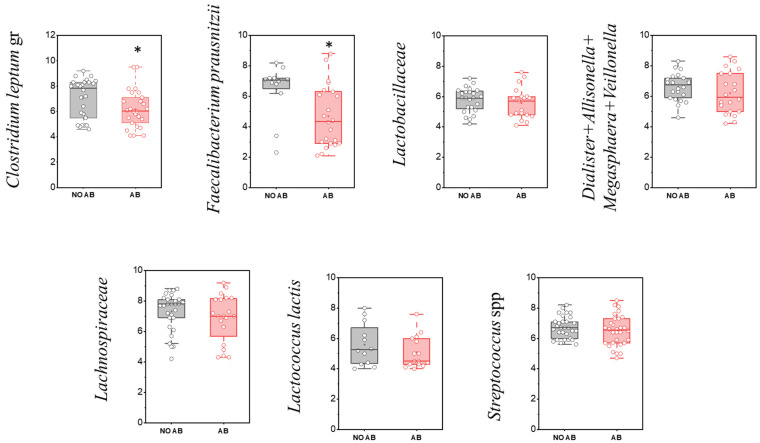
Comparison of quantitative indicators (GEs/g) of Firmicutes (Bacillota) representatives in children’s groups: without antibiotic therapy (no AB) and with antibiotic therapy (AB). Two sample *t* Test, * *p* < 0.05.

**Figure 7 antibiotics-14-01245-f007:**
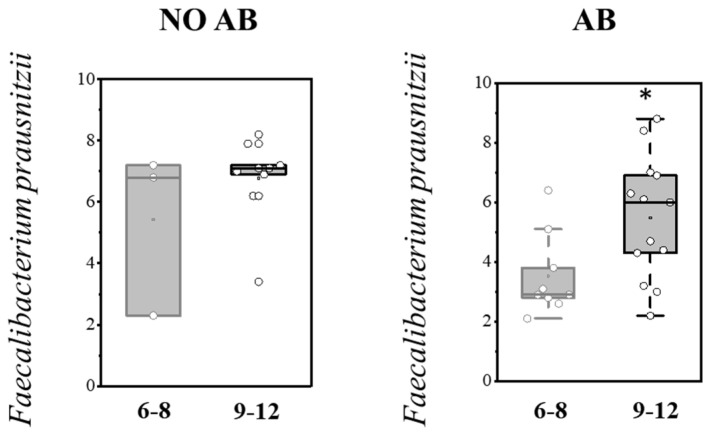
Analysis of quantitative indicators (Lg (GE/g)) of *F. prausnitzii* in the overall pediatric population depending on age (6–8 months and 9–12 months). Two sample *t* Test, * *p* < 0.05.

**Figure 8 antibiotics-14-01245-f008:**
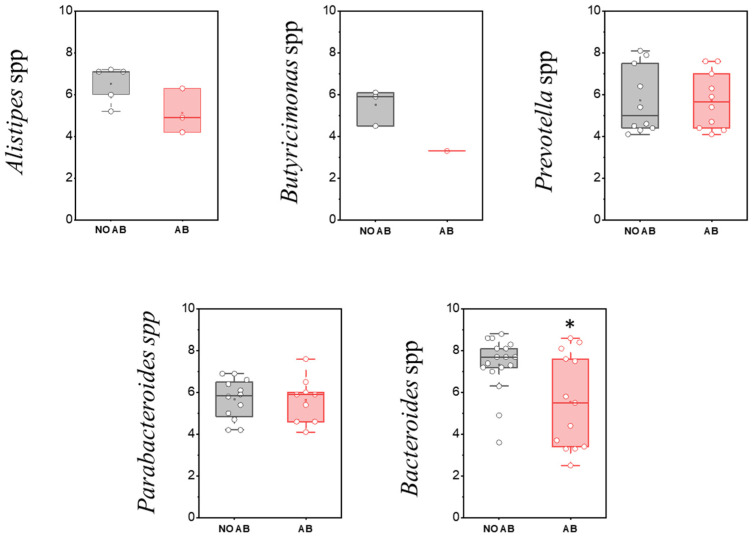
Analysis of quantitative indicators (GEs/g) of the Bacteroidota (formerly Bacteroidetes) phylum in children’s groups: without antibiotic therapy (no AB) and with antibiotic therapy (AB). Two sample *t* Test, * *p* < 0.05.

**Figure 9 antibiotics-14-01245-f009:**
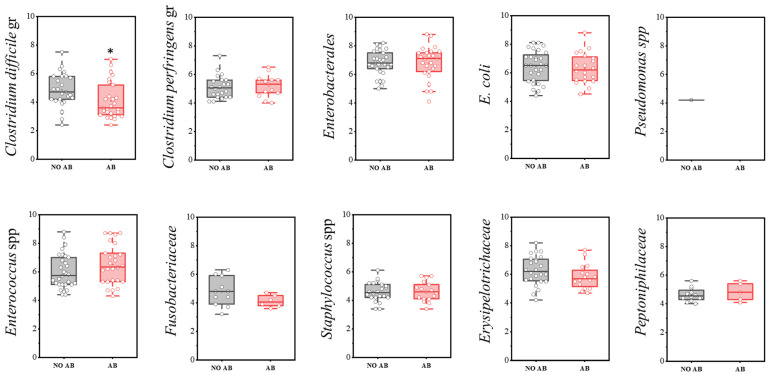
Analysis of quantitative indicators (CFU/g) of representatives of conditionally pathogenic microbiota in children’s groups: without antibiotics (no AB) and with antibiotic therapy (AB). Two sample *t* Test, * *p* < 0.05.

**Figure 10 antibiotics-14-01245-f010:**
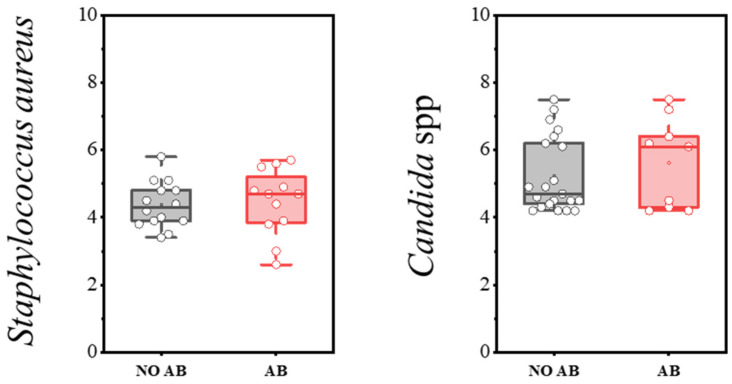
Analysis of quantitative indicators (log CFU/g) of *S. aureus* and *Candida* spp. in children’s groups: without antibiotics (NO AB) and with antibiotics therapy (AB).

**Table 1 antibiotics-14-01245-t001:** Fecal samples from healthy children aged 6 to 12 months (NO AB group).

No.	Gender	Date of Birth	ABT	Breastfeeding (Yes/No)	Complementary Foods (Yes/No)	Age (Months)
F1	M	13 June 2023	no	No	Yes	7
F2	M	18 June 2023	no	Yes	Yes	7
F3	M	13 June 2023	no	Yes	Yes	8
F9	F	18 September 2023	no	No	Yes	6
F10	M	2 September 2023	no	Yes	Yes	7
F11	F	23 June 2023	no	Yes	Yes	9
F12	M	6 May 2023	no	No	Yes	11
F13	F	5 May 2023	no	Yes	No	12
F16	M	15 May 2023	no	No	Yes	12
F17	M	2 October 2023	no	Yes	Yes	7
F21	F	7 May 2023	no	Yes	Yes	12
F23	F	5 October 2023	no	mix	Yes	7
F24	F	28 May 2023	no	No	Yes	12
F25	F	9 October 2023	no	Yes	Yes	7
F29	F	9 June 2023	no	Yes	Yes	12
F30	M	24 June 2023	no	No	Yes	11
F31	F	13 October 2023	no	Yes	Yes	8
F34	M	1 June 2023	no	Yes	Yes	12
F35	M	11 September 2023	no	Yes	No	9
F44	M	2 January 2024	no	Yes	Yes	7
F45	F	1 February 2024	no	No	Yes	6
F47	M	19 December 2023	no	No	Yes	8
F48	F	30 January 2024	no	No	Yes	7
F49	M	30 January 2024	no	No	Yes	7
F50	F	30 January 2024	no	No	Yes	7
F51	F	10 August 2023	no	Yes	Yes	12
F52	F	30 January 2024	no	No	Yes	7
F56	M	2 January 2024	no	Yes	Yes	8
F57	M	29 October 2023	no	No	Yes	10
F59	F	29 September 2023	no	No	Yes	11

**Table 2 antibiotics-14-01245-t002:** Fecal samples from children aged 6 to 12 months who received β-lactam antibiotic therapy (AB group).

No.	Gender	Date of Birth	ABT	Breastfeeding (Yes/No)	Complementary Foods (Yes/No)	Age (Months)
F4	M	17 March 2023	yes	No	No	12
F5	M	11 September 2023	yes	Yes	Yes	6
F6	F	5 May 2023	yes	Yes	Yes	11
F7	M	23 March 2023	yes	Yes	Yes	12
F8	M	3 October 2023	yes	No	Yes	6
F14	F	5 October 2023	yes	No	Yes	7
F15	M	12 June 2023	yes	No	Yes	11
F18	F	29 October 2023	yes	No	Yes	6
F19	M	21 September 2023	yes	No	Yes	8
F20	M	6 June 2023	yes	No	Yes	11
F22	M	11 December 2023	yes	Yes	No	6
F26	M	14 June 2023	yes	Yes	Yes	11
F27	M	13 October 2023	yes	No	Yes	8
F28	M	11 July 2023	yes	No	No	10
F32	F	1 December 2023	yes	Yes	No	6
F33	F	4 July 2023	yes	No	Yes	11
F36	M	12 July 2023	yes	No	Yes	11
F37	F	25 August 2023	yes	No	Yes	10
F38	M	26 September 2023	yes	No	Yes	8
F39	M	19 September 2023	yes	No	Yes	9
F41	F	31 October 2023	yes	No	Yes	8
F42	F	22 September 2023	yes	Yes	Yes	9
F43	M	5 November 2023	yes	No	Yes	8
F46	M	12 September 2023	yes	Yes	Yes	11
F53	M	20 November 2023	yes	Yes	Yes	9
F54	M	19 March 2024	yes	Yes	No	6
F55	M	12 September 2023	yes	No	Yes	12
F58	M	26 November 2023	yes	Yes	Yes	9
F60	M	2 October 2023	yes	Yes	Yes	11
F61	M	6 September 2023	yes	Yes	Yes	12

**Table 3 antibiotics-14-01245-t003:** Indices of species richness or diversity used in the study.

Indexes Alpha Diversity	Formula
Simpson	S = Σ (n*_i_*/N)^2^
Shannon	H’ = −Σ (p*_i_* × ln p*_i_*)

Note: N—sample size (community size); n*_i_*—the number of individuals of the *i*-th species; p*_i_*—the relative abundance of the 1st species (n*_i_*/N).

**Table 4 antibiotics-14-01245-t004:** Frequency of occurrence of metabolically active “infant” species of the genus *Bifidobacterium* in children’s groups: without antibiotic therapy (No AB) and with antibiotic therapy (AB).

Microorganisms	NO AB (*n* = 30)	Frequency (%)	AAB (*n* = 30)	Frequency (%)	Chi-Square (χ^2^)
*Bifidobacterium bifidum*	23	76.7	27	90.0	*p* > 0.05
*Bifidobacterium breve*	21	70.0	24	80.0	*p* > 0.05
*Bifidobacterium longum* subsp. *infantis*	15	50.0	14	46.7	*p* > 0.05
*Bifidobacterium longum* subsp. *longum*	17	56.7	18	60.0	*p* > 0.05

**Table 5 antibiotics-14-01245-t005:** Detection frequency of metabolically active “adult” *Bifidobacterium* species in children’s groups: without antibiotic therapy (No AB) and with antibiotic therapy (AB).

Microorganisms	NO AB Group (*n* = 30)	Frequency (%)	AB Group (*n* = 30)	Frequency (%)	Chi-Square (χ^2^)
*Bifidobacterium adolescentis*	7	23.3	9	30.0	*p* > 0.05
*Bifidobacterium animalis* subsp. *lactis*	13	43.3	13	43.3	*p* > 0.05
*Bifidobacterium catenulatum* ssp.	12	40.0	8	26.7	*p* > 0.05
*Bifidobacterium dentium*	12	40.0	8	26.7	*p* > 0.05

**Table 6 antibiotics-14-01245-t006:** Detection frequency of Bacillota (Firmicutes) representatives in children’s groups: without antibiotic therapy (No AB) and with antibiotic therapy (AB). Chi-squared test, * *p* < 0.05, ** *p* < 0.01.

Microorganisms	NO AB (*n* = 30)	Frequency (%)	AB (*n* = 30)	Frequency (%)	Chi-Square (χ^2^)
*Clostridium leptum gr*	26	86.7	26	86.7	*p* > 0.05
*Dialister + Allisonella + Megasphaera + Veillonella* **	29	96.7	20	66.7	0.003
*Enterococcus* spp.	30	100.0	26	86.7	*p* > 0.05
*Faecalibacterium prausnitzii* *	12	40.0	22	73.3	0.010
*Lachnospiraceae* *	27	90.0	20	66.7	0.029
*Lactobacillaceae*	24	80.0	19	63.3	*p* > 0.05
*Lactococcus lactis*	12	40.0	15	50.0	*p* > 0.05

**Table 7 antibiotics-14-01245-t007:** Age-related changes in the detection rate of *F. prausnitzii* in children’s groups: without antibiotic therapy (No AB) and with antibiotic therapy (AB).

Microorganisms	6–8 Months (*n* = 30)	Frequency (%)	9–12 Months (*n* = 30)	Frequency (%)	Chi-Square (χ^2^)
*F. prausnitzii* NO AB	9	30	13	43.3	*p* > 0.05
*F. prausnitzii* AB	3	10	9	30	*p* = 0.053

**Table 8 antibiotics-14-01245-t008:** Frequency of detection of Bacteroidota (formerly Bacteroidetes) representatives in groups of children: without antibiotic therapy (no AB) and with antibiotic therapy (AB).

Microorganisms	NO AB (*n* = 30)	Frequency (%)	AB (*n* = 30)	Frequency (%)	Chi-Square (χ^2^)
*Alistipes* spp.	5	16.7	3	10.0	*p* > 0.05
*Bacteroides* spp.	17	56.7	13	43.3	*p* > 0.05
*Butyricimonas* spp.	3	10.0	1	3.3	*p* > 0.05
*Parabacteroides* spp.	12	40.0	9	30.0	*p* > 0.05
*Prevotella* spp.	10	33.3	10	33.3	*p* > 0.05

**Table 9 antibiotics-14-01245-t009:** Frequency of detection of representatives of conditionally pathogenic microbiota in children’s groups: without antibiotic therapy (No ABT) and with antibiotic therapy (ABT). Chi-squared test, * *p* < 0.05.

Microorganisms	NO AB Group (*n* = 30)	Frequency (%)	AB Group (*n* = 30)	Frequency (%)	Chi-Square (χ^2^)
*Clostridium difficile gr*	25	83.3	27	90.0	*p* > 0.05
*Clostridium perfringens gr* *	22	73.3	13	43.3	0.019
*E. coli* *	28	93.3	20	66.7	0.010
*Enterobacterales*	29	96.7	25	83.3	*p* > 0.05
*Enterococcus* spp.	30	100.0	26	86.7	*p* > 0.05
*Erysipelotrichaceae* *	24	80.0	16	53.3	0.029
*Fusobacteriaceae*	11	36.7	6	20.0	*p* > 0.05
*Peptoniphilaceae **	12	40.0	5	16.7	0.045
*Prevotella* spp.	10	33.3	10	33.3	*p* > 0.05
*Pseudomonas* spp.	1	3.3	0	0.0	*p* > 0.05

**Table 10 antibiotics-14-01245-t010:** Frequency of *Staphylococcus aureus* detection in children’s groups: without antibiotics (NO AB) and with antibiotics (AB).

Microorganisms	NO AB Group (*n* = 30)	Frequency (%)	AB Group (*n* = 30)	Frequency (%)	Chi-Square (χ^2^)
*S. aureus*	14	47.0	12	40.0	*p* > 0.05
*Candida* spp.	12	40.0	9	30.0	*p* > 0.05

**Table 11 antibiotics-14-01245-t011:** OLS regression results highlighting factors influencing early childhood gut microbiota.

Variable	Coef	Std.Err	*p*-Value	Conf_Lower	Conf_Upper	Factor
ABT	−1.7908	0.9149	0.0552	−3.6229	0.0413	*Bacteroides* spp.
ABT	−1.5180	0.3502	0.0001	−2.2193	−0.8166	*Bifidobacterium* spp.
ABT	−2.4733	0.6541	0.0004	−3.7831	−1.1635	*Dialister Allisonella* *Megasphaera Veillonella*
ABT	−1.6585	0.7901	0.0402	−3.2406	−0.0764	*Coriobacteriia*
ABT	−2.0327	0.8016	0.0140	−3.6379	−0.4276	Lachnospiraceae
ABT	−1.7854	0.7157	0.0155	−3.2186	−0.3523	*Bifidobacterium catenulatum* ssp.
ABT	−2.0162	0.7346	0.0081	−3.4873	−0.5451	Erysipelotrichaceae
ABT	−1.8441	0.6830	0.0091	−3.2118	−0.4764	*E. coli*
ABT	−1.2229	0.4985	0.0172	−2.2211	−0.2246	Peptoniphilaceae
ABT	−1.5782	0.6544	0.0191	−2.8885	−0.2679	*Clostridium perfringens* gr
Breastfeeding	−1.6220	0.7758	0.0410	−3.1755	−0.0686	*Faecalibacterium prausnitzii*
Breastfeeding	−1.8040	0.5781	0.0028	−2.9617	−0.6464	*Akkermansia muciniphila*
Breastfeeding	−1.2088	0.6878	0.0842	−2.5860	0.1685	*Clostridium leptum* gr
Breastfeeding	−1.3544	0.7161	0.0637	−2.7883	0.0795	*Bifidobacterium catenulatum* ssp.
Breastfeeding	−1.7622	0.4988	0.0008	−2.7610	−0.7634	Peptoniphilaceae
Breastfeeding	−1.6065	0.6861	0.0227	−2.9803	−0.2327	*Bifidobacterium breve*
Breastfeeding	−0.5169	0.2894	0.0794	−1.0963	0.0626	*Streptococcus agalactiae*

## Data Availability

The original contributions presented in this study are included in the article/Appendix A. Further inquiries can be directed to the corresponding authors.

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
