# Peer review of "The Impact of Antimicrobial Therapy on the Development of Microbiota in Infants"

_antibiotics, 2025, doi:10.3390/antibiotics14121245_

Round 1

Reviewer 1 Report

Comments and Suggestions for Authors

The study compares microbiota diversity and colonization of key microorganisms between infants (born via spontaneous labor at ≥38 weeks) who received beta-lactam antibiotics and those who did not. The topic is highly relevant to pediatric health, microbiology, and immunology. Early-life gut microbiota is critical for overall health development, and antibiotics are widely prescribed in infancy.

The comparative analysis is focused on very specific age groups (6-8 and 9-12 months), labor type (spontaneous ≥38 weeks), and a specific antibiotic class (beta-lactams).
The introduction is clear, well-organized, and provides sufficient background for readers to understand the role of the gut microbiome in both children and adults. However, please specify the role of beta-lactam antibiotics in infants and clarify why the study focused on this class. Additionally, mention other antibiotic groups commonly prescribed in early childhood for context.

For Tables 1 and 2, please define the abbreviation ABP, as it has not been introduced previously. Consider adding an “Abbreviations” section at the end of the manuscript to systematically list all abbreviations used throughout the text.

Please also ensure that the in-text citations follow the journal’s formatting requirements. Finally, check that microorganism names are consistently italicized, for example, Faecalibacterium prausnitzii in line 465.

In my opinion, the main omission is the lack of discussion on potential confounding factors (e.g., diet, breastfeeding, maternal microbiota, environmental exposures) that could influence gut microbiota composition. It would be useful to include a new table or update an existing one detailing infants’ diets-breastfed versus formula-fed-and indicate which infants are already consuming solid foods. Diet has a major impact on gut microbiota and may contribute to misinterpretation of some results. Lines 130-134 are mainly descriptive and do not specify exactly what each infant is receiving; this section could be expanded with more discussion and comparisons to other studies.

Additionally, the Discussion section is currently too brief. It should link your results to existing literature and provide a more thorough interpretation. If it is easier, you may combine Results and Discussion, but in that format, the Discussion content should be strengthened.

I recommend including five points on future directions and recommendations, incorporating your personal perspective on the topic. The Conclusion should be more general, and some material for the Future Directions section can be drawn from there.

Author Response

Reviewer 1

Reviewer:

The study compares microbiota diversity and colonization of key microorganisms between infants (born via spontaneous labor at ≥38 weeks) who received beta-lactam antibiotics and those who did not. The topic is highly relevant to pediatric health, microbiology, and immunology. Early-life gut microbiota is critical for overall health development, and antibiotics are widely prescribed in infancy.

The comparative analysis is focused on very specific age groups (6-8 and 9-12 months), labor type (spontaneous ≥38 weeks), and a specific antibiotic class (beta-lactams).

The introduction is clear, well-organized, and provides sufficient background for readers to understand the role of the gut microbiome in both children and adults. However, please specify the role of beta-lactam antibiotics in infants and clarify why the study focused on this class. Additionally, mention other antibiotic groups commonly prescribed in early childhood for context.

Answer:

Thank you for your comment. The selection of β-lactam antibiotics in our study is justified by their high prevalence in pediatric practice, particularly among infants and young children. Epidemiological data indicate that these agents represent the most frequently prescribed class of antibiotics for treating bacterial infections of the respiratory tract, urinary tract infections, and other acute conditions.

Furthermore, β-lactam antibiotics possess broad-spectrum activity against both Gram-positive and Gram-negative bacteria and are readily accessible, which explains their preference in clinical settings. Consequently, examining their impact on the gut microbiota of children is especially relevant, since this class of antibiotics is considered among the safest options and is classified as first-line (empiric) therapy for early childhood infections. Their effects on the developing microbiome may have long-term clinical implications.

Penicillins, such as amoxicillin, are first-line agents for bacterial tonsillitis, pneumonia, and other infections in children, owing to their good tolerability and spectrum of activity. Cephalosporins, particularly third- and fourth-generation agents, carbapenems, are used in more severe cases or in cases of penicillin allergy, as well as for complicated infections such as sepsis or severe pneumonia.

Besides β-lactams, other antibiotic classes are widely used in pediatric practice, depending on the infection type, child's age, and the suspected pathogen. Macrolides, such as azithromycin and erythromycin, are frequently employed in respiratory diseases and in cases of allergy or intolerance to penicillins. They are especially effective against atypical pathogens, including Mycoplasma and Chlamydia, making them an important group in the treatment of pediatric infections.

For severe infections caused by Gram-negative bacteria, aminoglycosides such as amikacin and netilmicin are utilized. Given their increased toxicity risks to the kidneys and auditory system, especially in neonates and infants, their use requires careful dose management and blood level monitoring. Considering the high incidence of urinary system pathologies in our cohort, this group of antibiotics was not applied in our study.

In summary, antibiotic choice in children is driven by the need to ensure treatment efficacy, minimize adverse effects, and consider the child's specific physiological features. Investigating the effects of these different antibiotic classes on the microbiota must account for their diversity and mechanisms of action.

___________________________________________________________________________________

Reviewer:

For Tables 1 and 2, please define the abbreviation ABP, as it has not been introduced previously. Consider adding an “Abbreviations” section at the end of the manuscript to systematically list all abbreviations used throughout the text.

Answer:

We apologize. Indeed, the abbreviation in these tables should be corrected to ABT (Antibiotics Therapy)! Corrected! We also agree and have added the list of abbreviations at the end of the manuscript.

_____________________________________________________________________________________

Reviewer:

Please also ensure that the in-text citations follow the journal’s formatting requirements. Finally, check that microorganism names are consistently italicized, for example, Faecalibacterium prausnitzii in line 465.

Answer:

Checked and corrected!

_____________________________________________________________________________________

Reviewer:

In my opinion, the main omission is the lack of discussion on potential confounding factors (e.g., diet, breastfeeding, maternal microbiota, environmental exposures) that could influence gut microbiota composition. It would be useful to include a new table or update an existing one detailing infants’ diets-breastfed versus formula-fed-and indicate which infants are already consuming solid foods. Diet has a major impact on gut microbiota and may contribute to misinterpretation of some results. Lines 130-134 are mainly descriptive and do not specify exactly what each infant is receiving; this section could be expanded with more discussion and comparisons to other studies.

Answer:

Thank you for your attentive and constructive comments. You are absolutely right: considering potential confounding factors such as diet, environmental exposure, maternal microbiota status, and feeding regimen is important for a complete and accurate interpretation of our results. We agree that including additional information in the table — with a more detailed description of infant nutrition (including complementary feeding), feeding type (breastfeeding or artificial feeding), and nutritional features — will significantly enhance the analytical value of the work. Indeed, diet has a substantial impact on gut microbiota, and accounting for it is essential for a more precise analysis of the relationships. We have added all this additional information! Unfortunately, we did not analyze maternal microbiota in parallel, but this is planned for the near future.

Additionally, based on our available data, we have created supplementary graphs considering the factor of feeding type. Please review them. The statistically significant differences are no longer present in all taxonomic groups due to the reduction in sample size by nearly half, but the overall trend, as described in our manuscript, remains relevant.

_____________________________________________________________________________________

Reviewer:

Additionally, the Discussion section is currently too brief. It should link your results to existing literature and provide a more thorough interpretation. If it is easier, you may combine Results and Discussion, but in that format, the Discussion content should be strengthened.

I recommend including five points on future directions and recommendations, incorporating your personal perspective on the topic. The Conclusion should be more general, and some material for the Future Directions section can be drawn from there.

Answer:

The discussion has been completely corrected taking into account all comments.

_____________________________________________________________________________________

Reviewer 2 Report

Comments and Suggestions for Authors

This study provides an important examination of antibiotic-induced microbiota changes in a pediatric population. The qPCR approach to quantify microbial groups is a strength, and the findings of reduced diversity and beneficial taxa in antibiotic-exposed infants are valuable. However, to enhance the manuscript for publication, significant revisions are needed, particularly regarding the interpretation of results and acknowledging the study's context and limitations.

Major issues

  • Antibiotics were administered at different times (postnatally, at 3 months, and to outpatients at unknown times). The impact of antibiotics on the microbiota is known to be highly dependent on the timing of exposure.
  • The group consists of infants who were given antibiotics for different lengths of time, and for a wide range of clinical reasons (such as pneumonia, urinary tract infections, or post-surgery). The inflammatory nature of their underlying illnesses can also affect their microbiota, making it difficult to distinguish the impact of the antibiotics from the effects of the diseases themselves.
  • The authors correctly identified and controlled for the delivery mode by including only vaginally delivered infants. However, their handling of the feeding method is inadequate and represents a significant confounding factor. The literature clearly indicates that breastfeeding and formula feeding produce distinctly different gut microbiota ecosystems. Although the groups have similar totals for feeding type, the analysis does not account for this variable at the individual level. As a result, the observed differences between the two groups could be influenced or obscured by the underlying variation caused by the feeding method that has not been addressed. The statistical analysis must be re-run using feeding type as a covariate to isolate the effect of antibiotic exposure from the effect of diet.
  • The cross-sectional design of this study, sampling at a single time point between 6-12 months, is a fundamental limitation. It prevents us from determining the dynamics of microbiota disruption and recovery after antibiotic exposure. Without baseline microbiota profiles before treatment, we cannot definitively link observed differences at 6-12 months to antibiotic use. Infants requiring antibiotics for conditions like congenital pneumonia or UTI may have had different microbiota states beforehand, which could influence both infection susceptibility and later microbiota composition. Therefore, while the associations are valuable, they do not establish a direct causal relationship between antibiotic therapy and altered microbiota, reflecting a complex interplay between initial host factors, illness, and treatment.
  • The discussion section has critical flaws for two main reasons. First, it fails to place the findings within the context of existing scientific literature. Second, it does not acknowledge or address the study's limitations. Additionally, the section is descriptive rather than mechanistic. The authors could reference studies that explain why SCFA-producing bacteria are diminished, such as the fact that anaerobic bacteria are often more susceptible to broad-spectrum antibiotics and that β-lactams target cell wall synthesis.

Minor issues

  • The explanation of the Simpson and Shannon indices is repeated verbatim in the Methods and Results sections (Page 6). This is redundant and should be condensed.
  • In the abstract, the authors mention "a trend toward reduction" (e.g., for Lactobacillaceae). It is crucial to report the actual p-value for these analyses so that readers can assess the strength of the evidence.
Comments on the Quality of English Language

The English is generally clear and understandable, but it is not fine for publication in a reputable international journal. It requires thorough editing by a native English speaker or a professional scientific editing service.

For example, "nourishing flora" is better phrased as "nutritional flora" or "commensal flora"; "writing of the manuscript" is more natural as "manuscript preparation" or "writing – original draft".

Author Response

Main comments

Comments 1:

Antibiotics were administered at different times (postnatally, at 3 months, and to outpatients at unknown times). The impact of antibiotics on the microbiota is known to be highly dependent on the timing of exposure.

The group consists of infants who were given antibiotics for different lengths of time, and for a wide range of clinical reasons (such as pneumonia, urinary tract infections, or post-surgery). The inflammatory nature of their underlying illnesses can also affect their microbiota, making it difficult to distinguish the impact of the antibiotics from the effects of the diseases themselves.

Response 1:

Absolutely correct that the impact of antibiotics on the microbiota depends on the timing of their administration. In our case, most children received antibiotics during the postoperative period, that is, within the first month of life. We plan to conduct a more detailed analysis in the future, including studying the temporal dynamics of antibiotic effects at different time frames. However, this will require a significantly larger sample size.

Regarding the broad indications for prescribing antibiotics — we agree that various clinical situations and disease characteristics can independently influence the microbiota. In our sample, we tried to exclude gastrointestinal pathologies, as they are a major factor affecting gut microbiota. We believe that other pathologies have less obvious impacts, although they may influence it through various mechanisms. It is important to emphasize that prescribing antibiotics to children without valid medical reasons is prohibited, which excludes the possibility of assessing their effects in completely healthy children.

In this work, we do not consider the potential influence of inflammatory processes, and we acknowledge as a limitation that it is currently challenging to fully profile the impact of each specific clinical indication on the microbiota. This could be a promising direction for future research.

Comments 2:

The authors correctly identified and controlled for the delivery mode by including only vaginally delivered infants. However, their handling of the feeding method is inadequate and represents a significant confounding factor. The literature clearly indicates that breastfeeding and formula feeding produce distinctly different gut microbiota ecosystems. Although the groups have similar totals for feeding type, the analysis does not account for this variable at the individual level. As a result, the observed differences between the two groups could be influenced or obscured by the underlying variation caused by the feeding method that has not been addressed. The statistical analysis must be re-run using feeding type as a covariate to isolate the effect of antibiotic exposure from the effect of diet.

Response 2:

You are absolutely right: the mode of feeding plays a key role in shaping the child's microbiota and can significantly influence the analysis results. In our study, we did not account for this factor at the level of overall group distribution, but indeed, analyzing the individual impact of feeding type is very important and was not performed, which is a limitation on our part.

However, based on the information we have on overall nutrition, we have additionally prepared comparative data on the main taxa for comparison between groups of children by type of feeding. We are attaching these data.

It is clear that even with a smaller sample size, statistically significant differences remain in some taxa, which might indicate the contribution of antibiotics to microbiota formation in both groups. Especially in the group with artificial feeding, the difference is more pronounced, suggesting the importance of breastfeeding—including its role in the formation and strengthening of the gut microbiota's protective function in newborns.

In the future, we plan to expand the analysis and consider this important factor in more detail. Additionally, if the information proves to be of interest, we can include it as supplementary material. Thank you for your constructive recommendation; it will be included into our plans for improving the research.

Comments 3:

The cross-sectional design of this study, sampling at a single time point between 6-12 months, is a fundamental limitation. It prevents us from determining the dynamics of microbiota disruption and recovery after antibiotic exposure. Without baseline microbiota profiles before treatment, we cannot definitively link observed differences at 6-12 months to antibiotic use. Infants requiring antibiotics for conditions like congenital pneumonia or UTI may have had different microbiota states beforehand, which could influence both infection susceptibility and later microbiota composition. Therefore, while the associations are valuable, they do not establish a direct causal relationship between antibiotic therapy and altered microbiota, reflecting a complex interplay between initial host factors, illness, and treatment.

Response 3:

You are correct: the design of our study indeed limits the ability to track microbiota dynamics over time and to understand how it recovers after antibiotic therapy. However, our primary goal was to understand the establishment of the microbiota during the first year of life and its changes in response to antibiotic treatment.

Although we lack microbiota data prior to treatment initiation, which prevents us from definitively linking observed differences solely to antibiotics, our main focus was not on analyzing the impact of antibiotics on microbiota at individual points (the negative effects of antibiotics are already well known). Instead, we aimed to assess overall patterns and correlations.

It is also true that different pre-existing microbiota states in infants—such as those caused by congenital pneumonia—could influence subsequent microbiota composition. This important aspect indicates that our analysis is more correlational than causative. To clarify causal relationships in future research, it would be necessary to include baseline data collected before therapy and to conduct longitudinal monitoring of microbiota changes over time. This could be one of the next stages in our research plans.

Comments 4:

The discussion section has critical flaws for two main reasons. First, it fails to place the findings within the context of existing scientific literature. Second, it does not acknowledge or address the study's limitations. Additionally, the section is descriptive rather than mechanistic. The authors could reference studies that explain why SCFA-producing bacteria are diminished, such as the fact that anaerobic bacteria are often more susceptible to broad-spectrum antibiotics and that β-lactams target cell wall synthesis.

Response 4:

The discussion has been completely corrected taking into account all comments.

_____________________________________________________________________________________

Minor corrections:

Comments 5:

The explanation of the Simpson and Shannon indices is repeated verbatim in the Methods and Results sections (Page 6). This is redundant and should be condensed.

Response 5:

Checked, deleted!

Comments 6:

In the abstract, the authors mention "a trend toward reduction" (e.g., for Lactobacillaceae). It is crucial to report the actual p-value for these analyses so that readers can assess the strength of the evidence.

Response 6:

We can add it, no problem, but we think that the p-value of 0.16 (t-test) is not very informative. However, in our subjective opinion, there is a trend of decrease visible on the graph.

Round 2

Reviewer 1 Report

Comments and Suggestions for Authors

Once again, I would like to congratulate the authors on the current and interesting topic. The new information added definitely contributes to the improvement of the entire manuscript.

In my opinion, the answer to my question regarding the choice of antibiotics should be included in the introduction. The authors have justified their choice of antibiotics very well, and in my opinion, this will sound good as part of the introduction. Your answer - from "The selection of β-lactam antibiotics in our study is justified by their high prevalence in pediatric practice, particularly among infants and young children. Epidemiological data indicate that these agents represent the most frequently prescribed class of antibiotics for treating bacterial infections of the respiratory tract, urinary tract infections, and other acute conditions....to....In summary, the choice of antibiotics in children is driven by the need to ensure treatment efficacy, minimize adverse effects, and consider the child's specific physiological characteristics. Investigating the effects of these different antibiotic classes on the microbiota must account for their diversity and mechanisms of action." I suggest you include it in the Introduction of the manuscript.

I noticed that the citations in the text are not formatted correctly, but the authors have not corrected them. Please refer to the journal's citation requirements.
"In-text citations
Place the reference number in square brackets before the punctuation, e.g., [1] or [1–3].
To cite a specific page, use a format like [5] (p. 10) or [6] (pp. 101–105)."

If you make these corrections, the article will be improved. With them, it can be accepted for publication.

Author Response

Сomments1

Once again, I would like to congratulate the authors on the current and interesting topic. The new information added definitely contributes to the improvement of the entire manuscript.

In my opinion, the answer to my question regarding the choice of antibiotics should be included in the introduction. The authors have justified their choice of antibiotics very well, and in my opinion, this will sound good as part of the introduction. Your answer - from "The selection of β-lactam antibiotics in our study is justified by their high prevalence in pediatric practice, particularly among infants and young children. Epidemiological data indicate that these agents represent the most frequently prescribed class of antibiotics for treating bacterial infections of the respiratory tract, urinary tract infections, and other acute conditions....to....In summary, the choice of antibiotics in children is driven by the need to ensure treatment efficacy, minimize adverse effects, and consider the child's specific physiological characteristics. Investigating the effects of these different antibiotic classes on the microbiota must account for their diversity and mechanisms of action." I suggest you include it in the Introduction of the manuscript.

Response 1: Thanks!  Added information about the choice of AB to the introduction!

Comments2: I noticed that the citations in the text are not formatted correctly, but the authors have not corrected them. Please refer to the journal's citation requirements.
"In-text citations
Place the reference number in square brackets before the punctuation, e.g., [1] or [1–3].
To cite a specific page, use a format like [5] (p. 10) or [6] (pp. 101–105)."

Response2:

  • All citations have been corrected!
  • In additionally English was corrected.

Thanks for your review! Best regards

Reviewer 2 Report

Comments and Suggestions for Authors

1) The supplementary ppt and descriptive comparisons do not address my request to include the feeding method as a covariate. The main results still lack any multivariate statistical analysis that utilizes feeding type as a covariate to separate the effects of antibiotic exposure from those of diet. It is essential to incorporate feeding into the statistical models—whether using regression or differential abundance methods—and to fully report the coefficients, CIs, and p-values, as I previously requested.  Without this inclusion, significant confounding remains, and the findings cannot be accurately interpreted as reflecting the effects of antibiotic exposure.

2) In the revised discussion, the sentence “Numerous studies confirm that improper or excessive use of antibiotics can disturb microbial communities in children's intestines…” is followed by only one citation (Arrieta et al., 2014). Please provide additional references to support the statement “numerous studies.

3) The English language issues previously noted have not been corrected in the updated manuscript. The expressions “nourishing flora” and “writing of the manuscript” still appear unchanged in the revised version.

Comments on the Quality of English Language

The English is generally clear and understandable, but it is not fine for publication in a reputable international journal. It requires thorough editing by a native English speaker or a professional scientific editing service.

For example, "nourishing flora" is better phrased as "nutritional flora" or "commensal flora"; "writing of the manuscript" is more natural as "manuscript preparation" or "writing – original draft".

Author Response

First of all, Thank you for your interest in our work and your deep understanding of the problem!

comments 1: The supplementary ppt and descriptive comparisons do not address my request to include the feeding method as a covariate. The main results still lack any multivariate statistical analysis that utilizes feeding type as a covariate to separate the effects of antibiotic exposure from those of diet. It is essential to incorporate feeding into the statistical models—whether using regression or differential abundance methods—and to fully report the coefficients, CIs, and p-values, as I previously requested.  Without this inclusion, significant confounding remains, and the findings cannot be accurately interpreted as reflecting the effects of antibiotic exposure.

response 1:

We conducted an analysis for each of the 36 factors, building separate multiple linear regression models. Predictors included factors such as antibiotic therapy and breastfeeding. Regression analyses were performed using the «statsmodels»( Seabold, Skipper, and Josef Perktold. “statsmodels: Econometric and statistical modeling with python.” Proceedings of the 9th Python in Science Conference. 2010 ) library, resulting in detailed outputs that include coefficient estimates, their statistical significance, and model quality metrics.

The analysis showed that ABT (antibiotic therapy) influences parameters such as:
Bifidobacterium_spp p = 0.00006
Bifidobacterium_catenulatum_ssp p = 0.0155
Dialister_Allisonella_Megasphaera_Veillonella p = 0.000376
Coriobacteriia p = 0.0402
Lachnospiraceae p = 0.013981
Erysipelotrichaceae p = 0.00809
E. coli p = 0.0091
Peptoniphilaceae p = 0.017
Clostridium_perfringens_gr p = 0.019

Parameters close to statistical significance (factor is ABT):
Bacteroides_spp p = 0.055
Enterobacterales p = 0.11
Lactobacillaceae p = 0.11
Fusobacteriaceae p = 0.1

Contribution of breastfeeding:
Staphylococcus_aureus p = 0.047
Faecalibacterium_prausnitzii p = 0.04
Akkermansia_muciniphila p = 0.0028
Peptoniphilaceae p = 0.0008

Parameters close to statistical significance (factor is breastfeeding):
Clostridioides_difficile p = 0.063202
Clostridium_leptum_gr p = 0.084
Bifidobacterium_catenulatum_ssp p = 0.064
Bifidobacterium_breve p = 0.022
Streptococcus_agalactiae p = 0.079

All data are included in the doc file (correlation.docx )

and cloud 

https://disk.yandex.ru/d/SbqUCV6s-K5jDA

comments 2: In the revised discussion, the sentence “Numerous studies confirm that improper or excessive use of antibiotics can disturb microbial communities in children's intestines…” is followed by only one citation (Arrieta et al., 2014). Please provide additional references to support the statement “numerous studies.

response 2: added

comments 3: The English language issues previously noted have not been corrected in the updated manuscript. The expressions “nourishing flora” and “writing of the manuscript” still appear unchanged in the revised version.

response 3: English was corrected

Round 3

Reviewer 2 Report

Comments and Suggestions for Authors

Dear Authors,

I appreciate that you have performed the multiple regression analyses, including the feeding method as a covariate. This is a crucial step. However, the key results of these analyses must be integrated into the main text, specifically the Results section. Burying these essential findings in a separate document prevents readers from fully evaluating the study’s conclusions. Please incorporate a summary of the significant results (e.g., in a table) into the manuscript, with the full outputs available in the Supplementary Materials.

Regarding the point on Lactobacillaceae, I understand your perspective. However, the standard in scientific reporting is to provide the statistical values that underpin descriptive terms like "trend." This allows readers to make their own assessment. A p-value of 0.16 indicates a lack of statistical significance. Labeling this as a "trend" in the abstract without the qualifying statistical context is misleading and overstates the finding. The abstract must accurately reflect the results.

Comments on the Quality of English Language

The English is generally clear and understandable, but it is not fine for publication in a reputable international journal. It requires thorough editing by a native English speaker or a professional scientific editing service.

For example, "nourishing flora" is better phrased as "nutritional flora" or "commensal flora"; "writing of the manuscript" is more natural as "manuscript preparation" or "writing – original draft".

Author Response

comments 1:

Dear Authors,

I appreciate that you have performed the multiple regression analyses, including the feeding method as a covariate. This is a crucial step. However, the key results of these analyses must be integrated into the main text, specifically the Results section. Burying these essential findings in a separate document prevents readers from fully evaluating the study’s conclusions. Please incorporate a summary of the significant results (e.g., in a table) into the manuscript, with the full outputs available in the Supplementary Materials.

response1:

Thank you for your feedback! We've added relevant analysis information to the results section (Multiple Regression Analysis and table 11) and additional materials to the "OLS_Regression_Results.xlsx" file.

comments 2:

Regarding the point on Lactobacillaceae, I understand your perspective. However, the standard in scientific reporting is to provide the statistical values that underpin descriptive terms like "trend." This allows readers to make their own assessment. A p-value of 0.16 indicates a lack of statistical significance. Labeling this as a "trend" in the abstract without the qualifying statistical context is misleading and overstates the finding. The abstract must accurately reflect the results.

reponse2:

Thank you for your understanding!

We fully agree with your opinion and that all data should be supported by statistical evidence! We have removed this phrase from the abstract to avoid future misunderstandings.  The results are described in more detail (including the p-value) in the relevant section (p. 14), and anyone interested can find this information.

Best regards!

Round 4

Reviewer 2 Report

Comments and Suggestions for Authors

no further comments

Comments on the Quality of English Language

The English is generally clear and understandable, but it is not fine for publication in a reputable international journal. It requires thorough editing by a native English speaker or a professional scientific editing service.

For example, "nourishing flora" is better phrased as "nutritional flora" or "commensal flora"; "writing of the manuscript" is more natural as "manuscript preparation" or "writing – original draft".